# Modeling of the In Vitro Release Kinetics of Sonosensitive Targeted Liposomes

**DOI:** 10.3390/biomedicines10123139

**Published:** 2022-12-05

**Authors:** Zeyad AlMajed, Najla M. Salkho, Hana Sulieman, Ghaleb A. Husseini

**Affiliations:** 1Biomedical Engineering Program, College of Engineering, American University of Sharjah, Sharjah P.O. Box 26666, United Arab Emirates; 2Department of Chemical Engineering, College of Engineering, American University of Sharjah, Sharjah P.O. Box 26666, United Arab Emirates; 3Materials Science and Engineering Program, College of Arts and Sciences, American University of Sharjah, Sharjah P.O. Box 26666, United Arab Emirates; 4Department of Mathematics and Statistics, College of Arts and Science, American University of Sharjah, Sharjah P.O. Box 26666, United Arab Emirates

**Keywords:** drug delivery systems, cancer treatment, liposomes, ligand targeting, ultrasound, drug release, kinetic model fitting

## Abstract

Targeted liposomes triggered by ultrasound are a promising drug delivery system as they potentially improve the clinical outcomes of chemotherapy while reducing associated side effects. In this work, a comprehensive model fitting was performed for a large dataset of liposomal release profiles with seven targeting moieties (albumin, cRGD, estrone, hyaluronic acid, Herceptin, lactobionic acid, and transferrin) in addition to the control liposomes under ultrasound release protocols. Two levels of ultrasound frequencies were tested: low frequency (20 kHz) at 6.2, 9, and 10 mW/cm^2^ as well as high frequencies (1.07 MHz and 3 MHz) at 10.5 and 173 W/cm^2^. At a low frequency, Hixson–Crowell, Korsmeyer–Peppas, Gompertz, Weibull, and Lu–Hagen showed good fits to our release profiles at all three power densities. At high frequencies, the former three models reflected the best fit. These models will aid in predicting drug release profiles for future in vitro studies.

## 1. Introduction

Cancer is one of the leading causes of death worldwide, with nearly 10 million death cases reported in 2020, according to the World Health Organization [1]. Among the various cancer treatment modalities, chemotherapy is commonly used as either a primary treatment or administered after/before other primary treatments (e.g., surgery), with the latter being known as adjuvant and neoadjuvant therapies. Depending on many factors, including the extent of malignancy and the efficacy of the cytotoxic drug, chemotherapy can play different roles, such as curing cancer with suppressed recurrence, controlling cancer to prolong patients’ survival when a complete cure is not possible, and relieving symptoms to improve patients’ quality of life [2].

Chemotherapy inhibits the growth and proliferation of cancer cells by interfering with the phases of the cell cycle. Cytotoxic chemotherapy targets all the fast-growing cells, including normal and malignant cells, which results in many side effects such as hair loss, nausea, vomiting, and various organ dysfunction. In addition, the tumor could be intrinsically resistant to chemotherapeutic drugs before the course of treatment or could acquire resistance after treatment, thus rendering the drugs ineffective. Drug resistance is believed to be the leading cause of treatment failure in over 90% of patients with metastatic cancer [3]. One approach employed clinically to evade drug resistance is to use a cocktail of chemotherapeutic drugs, such as the combination of mechlorethamine, vincristine, procarbazine, and prednisone (MOPP) for Hodgkin’s lymphoma [4]; cyclophosphamide, doxorubicin, vincristine, and prednisone (CHOP) for non-Hodgkin lymphoma [5]; docetaxel and cyclophosphamide (TC) for breast cancer [6]; and capecitabine and oxaliplatin (CAPOX) and folinic acid, fluorouracil, and oxaliplatin (FOLFOX) for colon cancer [7] and others. The combination of chemotherapeutics results in a synergetic effect, thus making the anticancer response more potent overall. However, it is important to note that some chemotherapy regimens may also have serious side effects that can be life-threatening due to incompatibility issues (i.e., multiple drug interference). Another approach to reduce drug resistance and the overall side effects of chemotherapy is to encapsulate drugs in nanocarriers that selectively target the tumor without affecting healthy tissues [8].

Nanocarriers used in drug delivery systems (DDSs) provide drug storage and allow for spatiotemporal release of their payload upon stimulation. Those nanosized vehicles can be conjugated to a biomarker in order to identify a specific tumor receptor. Liposomes are one of the attractive nanocarriers used frequently in DDSs. They are spherical nanocapsules composed of phospholipid bilayers with sizes ranging from 20 to 1000 nm in diameter. Liposomes can accommodate hydrophilic and hydrophobic drugs in their core and shell bilayer, respectively. Their surface can be functionalized with drugs and ligands that act as biomarkers for tumor identification and treatment. In addition, liposomes can be synthesized to respond to a desired stimulus/trigger, such as pH, enzymes, ultrasound, light, etc. [9,10,11,12]. Tissue penetration, selective energy deposition, and cost are important considerations when selecting such a stimulus. Magnetic fields, light, and microwaves have been considered as triggers but are limited to surface tumors [13,14] because X-rays are ionizing, and radiofrequency is invasive. Table 1 shows the advantages and disadvantages of drug delivery trigger mechanisms. Ultrasound (US) has significant advantages over other triggers since it is non-ionizing, with an established synergistic effect when utilized in conjunction with chemotherapeutic agents. In our laboratory, we are interested in synthesizing sonosensitive liposomes with several targeting moieties conjugated to their surface. These targeting moieties act as biomarkers to identify receptors that are overexpressed only on certain types of tumors.

This work aims to examine the kinetics of drug release from sonosensitive liposomes with several targeting moieties based on data gathered by [15,16,17,18,19,20,21,22,23,24]. The release experiments were conducted in vitro. An enormous dataset of nearly two million data points was processed in MATLAB to fit the drug release profiles to various kinetic models well-established in the literature for DDSs. By identifying the kinetic models with the best fit for our data, release profiles can be predicted for future in vitro studies. The details of the targeting moieties conjugated to liposomes used in this study are summarized in Table 2.

## 2. Results and Discussion

### 2.1. Low-Frequency Data Fitting

The cumulative fractional drug release (*CFR*) of seven moieties and their controls were fitted to the models reported in Table 3 at 20 kHz and three power densities with nine replicates per moiety per power density. MATLAB executed the model fittings, and the results were compiled in an excel sheet reporting the sum of the squared errors (*SSE*) values with the estimated coefficient(s) for each model. The *SSE* measures the amount of the response variability that the fitted model fails to capture. Lower SEE values indicate a better fit. Statistical analysis was performed using Minitab^®^, and the results were displayed as heatmaps for qualitative comparison, as shown in Figure 1. The accuracy of the model fitting in the heatmaps is based on the logarithm base 10 of the *SSE* values since *SSE* values are small ~10^−3^. The scale in the heatmaps is rated by a color gradient from blue to red, with the former indicating a better fit (less *SSE*). Furthermore, according to Figure 1, a few models reflected good fitting to our LF release data: Hixson–Crowell, Korsmeyer–Peppas, Weibull, Gompertz, and Lu–Hagen.

The low-frequency ultrasound (LFUS) release experiments were conducted with ON–OFF cycles in order to minimize the thermal effect on release profiles. Based on Figure 1, it is observed that the Lu–Hagen model performs better than the other models (if we exclude the polynomial fit) at the lowest power density of 10 mW/cm^2^, where thermal effects are smaller compared to other power densities, yet still measurable. In addition, and as expected, the Lu–Hagen model fits the data better when thermal effects are concentrated. This can be achieved by the continuous application of US, an extended duration of the ON-cycles, or using elevated power densities. In fact, the accuracy of the Lu–Hagen model increased for the power densities of 9 and 10 mW/cm^2^ compared to 6.2 mW/cm^2^ due to amplified thermal effects, as shown in Figure 1. Among all the models, a simple fitting of the release data to a polynomial of order three performed the best for all the power densities; however, it lacks the physical interpretation of the estimated coefficients.

The Korsmeyer–Peppas model also reflected a suitable fit for all the power densities, as shown in the heatmap. The estimated coefficients are reported in Table 4. Coefficients *a* and *b* correspond to *k_kp_* and n of the Korsmeyer–Peppas model, as shown in Table 3. The range for coefficient *b* (exponent of the model) spans between 0.62 and 0.99 (excluding estrone), thus implying a non-Fickian transport. Anomalous transport is a type of non-Fickian transport with 0.43 < *b* < 0.85 for a spherical geometry (such as liposomes) [35]. Most of our moieties follow anomalous transport, with very few expressing super case II transport (*b* > 0.85). Anomalous transport is characterized by solvent diffusion and relaxation of polymeric chains occurring with comparable magnitudes [35]. Thus, drug release is driven by diffusion and swelling. When the velocity of solvent diffusion is higher than polymeric relaxation, the mechanism of drug release is characterized by super case II transport.

Weibull, an empirical model, showed a good fit for our *CFR* data at all power densities. However, the parameters in the equations do not describe the mechanism of drug release. Those parameters only reflect the timescale and the shape of the *CFR* curve. The curve shape can be exponential (*b* ≤ 1) or sigmoid with an inflection point (*b* > 1) [35]. Around 52.4% of our moieties showed parabolic release profiles, while the remaining ones reflected a sigmoidal shape according to the estimated coefficient of the exponent. Figure 2 depicts the best-fitting models for albumin-conjugated liposomes at 6.2 mW/cm^2^ using the batches with the lowest *SSE* values.

Both Higuchi and Baker–Lonsdale models resulted in poor fitting to all power densities at low frequency. Higuchi’s model was originally developed for planar matrices with unidirectional diffusion, which is not representative of our spherical nanoparticles. In addition, the diffusivity of the drug is assumed to be constant, which is valid only for matrices with negligible change in their dimension due to swelling or dissolution. The Baker–Lonsdale model was developed based on Higuchi’s model and showed a poor fit as well.

### 2.2. High-Frequency Data Fitting

Release experiments were conducted at two levels of high-frequency ultrasound (HFUS): 1.07 MHz (10.5 W/cm^2^) and 3.00 MHz (173 W/cm^2^). Heatmaps with a log_10_
*SSE* scale were generated for three types of targeting moieties, in addition, to control liposomes, as shown in Figure 3. Models representing the best fits to albumin and transferrin-conjugated liposomes in addition to control liposomes are the Hixson–Crowell, Korsmeyer–Peppas, and Gompertz, and similar to the low-frequency results, a polynomial of order three showed an excellent fit for high-frequency release profiles. The experiments conducted at high frequency included fewer replicates per targeting moiety per power density and a longer acquisition time per measurement. Therefore, it is premature to draw conclusions regarding the high-frequency modeling output at this point due to the relatively sparse dataset, compared to the low-frequency studies.

## 3. Materials and Methods

This section presents an overview of the materials, synthesis protocol of control liposomes, and the data acquisition, processing, and programming codes methodology. In addition, the kinetic models of drug release from targeted liposomes are summarized.

### 3.1. Materials

1,2-dipalmitoyl-sn-glycero-3-phosphocholine (DPPC) and 1,2-distearoyl-sn-glycero-3-phosphoethanolamine-*N*-[amino (polyethylene glycol)-2000] (ammonium salt) (DSPE-PEG_2000_-NH_2_) were obtained from Avanti Polar Lipids Inc. (Alabaster, AL, USA). 2,4,6 trichloro-1,3,5 triazine (cyanuric chloride (NCCl)_3_), cholesterol, estrone (ES), human holo-transferrin (T_f_), human serum albumin (HSA) (MW: 68 kDa), hyaluronic acid (HA) (MW: 170 kDa), N-Ethyl-N-(3-dimethylaminopropyl)carbodiimide (EDC), N-Hydroxysuccinimide (NHS), 2-(N-Morpholino) ethanesulfonic acid hemisodium salt (MES), Triton™ X-100, calcein disodium salt, ammonium sulfate salt, and Sephadex^®^ G-25 and G-100 were purchased from Sigma-Aldrich Chemie GmbH (Schnelldorf, Germany). Chloroform was obtained from Panreac Quimica S.A. (Barcelona, Spain). cRGD was obtained from Musechem (Fairfield, NJ, USA). Triethylamine (TEA) was obtained from Reidel-de Haën (Hanover, Germany). Doxorubicin-hydrochloride (Dox) was obtained from Euroasian Transcontinental (Mumbai, India). Herceptin was obtained from Hoffmann-La Roche Limited (Basel, Switzerland).

### 3.2. Synthesis of Control Liposomes

Non-targeted control liposomes encapsulating calcein were synthesized according to the thin-film hydration method. Briefly, DPPC, cholesterol, and DSPE-PEG_2000_-NH_2_ were dissolved in 3 to 4 mL chloroform in a round bottom flask at a molar ratio of 65:30:5, respectively. The flask was then placed in a rotary evaporator at 50 °C under vacuum for 20 min to evaporate the chloroform and obtain a greasy-like film deposited on the flask wall. The lipidic film was then hydrated with 2 mL of calcein (30 mM, pH = 7.4) dissolved in phosphate-buffered saline (PBS). The hydration was performed in a rotary evaporator at 60 °C for 45 min (without vacuum). The sample was sonicated in a 35-kHz sonication bath (Elma D-78224, Melrose Park, IL, USA) at 60 °C for 2 min to form unilamellar liposomes, after which the sample was extruded 30 times at 60 °C through the Avanti^®^ Mini-extruder with 200-nm polycarbonate filters (Avanti Polar Lipids, Inc., Alabaster, AL, USA). Liposomes were purified through the gel filtration method using Sephadex^®^ G-100 (equilibrated with PBS at pH = 7.4) to remove unencapsulated calcein.

The protocol mentioned above was followed to prepare Dox-encapsulated liposomes with minor modifications. Briefly, the thin film was hydrated with ammonium sulfate dissolved in distilled water (0.11 M, pH = 5.3–5.6) instead of calcein. In addition, excess ammonium sulfate was removed using Sephadex^®^ G-25 equilibrated with HEPES (0.26 M Sucrose, 0.005 M ascorbic acid, and 0.016 M HEPES). Subsequently, Dox dissolved in HEPES buffer (1:6 (*w*/*w*) Dox/lipid) was incubated with liposomes in a water bath at 60 °C for 45 min with mild stirring. Finally, the liposomal solution was centrifuged through a Sephadex^®^ G-25 column equilibrated with PBS (pH = 7.4).

Depending on the targeting moiety (Table 2), the conjugation to liposomes was performed either by reacting the lipids with the targeting moiety before the synthesis of liposomes or targeting was accomplished after the synthesis of liposomes. For the detailed protocols, refer to [15,16,17,18,19,20,21,22,23,24].

### 3.3. Data Acquisition

Experiments were conducted in vitro to measure drug release from 7 targeted liposomes (Table 2) under low and high ultrasonic frequency levels. The LFUS was set at 20 kHz with power densities of 6.2, 9, and 10 mW/cm^2^, respectively. The HFUS experiments were conducted at 1.07 MHz (10.5 and 50 W/cm^2^) and 3 MHz (173 W/cm^2^). The power densities of the low-frequency ultrasonic probe (model VCX750, Sonics & Materials Inc., Newton, CT, USA) were determined by a hydrophone (Bruel & Kjaer 8103, Narum, Denmark).

The LFUS experiments were conducted using QuantaMaster QM 30 phosphorescence spectrofluorometer (Photon Technology International, Edison, NJ, USA). Samples were prepared by diluting 75 µL of liposomes in 3 mL of phosphate-buffered saline (PBS) inside a fluorescence cuvette. The cuvette was placed in the testing chamber of the spectrofluorometer where fluorescence was monitored. For calcein-loaded liposomes, the excitation and emission wavelengths were set at 495 and 515 nm, respectively. While for Dox-loaded liposomes, the excitation and emission wavelengths were set at 485 and 595 nm. A baseline was acquired before sonication by recording the initial fluorescence intensity $I_{o}\;$ for 60 s. Subsequently, ultrasound was applied in a pulsed mode: 20 s ON and 10 s OFF for calcein-loaded liposomes, as shown in Figure 4. For Dox-loaded liposomes, the OFF cycle was extended to 20 s instead of 10 s. The sonication was terminated when the fluorescence reached a plateau indicating no extra drug release by US. Next, liposomes were lysed by adding Triton X-100 (Tx100) to release all their content as characterized by the maximum fluorescence intensity $\;I_{\infty }$. The *CFR* of the drug was then calculated using the instantaneous intensity $\;I_{t}$, as follows:(1)$$CFR=\frac{I_{t}-I_{o}}{I_{\infty }-I_{o}}$$

### 3.4. Data Formatting and Processing

Raw data acquired from US release experiments were formatted into a specific structure in excel sheets to allow streamlined parsing and processing by MATLAB. The *CFR* of the drug at each time point was calculated and processed further in MATLAB to eliminate the OFF segments shown in Figure 4. As mentioned earlier, release experiments were performed using pulsed ultrasound to avoid the temperature rise by thermal effects, which interferes with the fluorescence (see Figure 4). However, the data should be continuous for model fitting, and thus all the OFF segments were trimmed from the release profiles using MATLAB (See Figure 5).

### 3.5. Kinetic Models for Drug Release and Error Estimation

In this section, drug release kinetic models are summarized. These models describe the drug dissolution profiles under certain conditions or assumptions. In order to use them in our data fitting, each model is rearranged first to yield an expression for the *CFR*, as shown in Table 3. In addition, regression fitting using a third-order polynomial was used but without a specific physical interpretation for the fitted parameters:(2)$$CFR=p_{1}t^{3}+p_{2}t^{2}+p_{3}t+p_{4}$$

The degree of variance between our *CFR* dataset ($y_{i}$) and those predicted by the kinetic models (${\hat{y}}_{i}$) is assessed using the sum of squared errors (*SSE*) with a smaller *SSE* value indicating a better fit:(3)$$SSE=\sum_{i=1}^{n}{\left(y_{i}-{\hat{y}}_{i}\right)}^{2}$$
where *n* is the number of data points.

### 3.6. MATLAB Codes

Codes were written in MATLAB to enable automated parsing and preprocessing of a huge dataset for more than 2 million data points saved in spreadsheets. The built-in fit functions available in MATLAB were used, which further simplified the fitting codes for nonlinear models. The release profiles (for control and targeted liposomes (Table 2)) were fitted to the kinetic models presented in Table 3 in their explicit form *CFR* = *f*(*t*) except for the Hixson–Crowell and Baker–Lonsdale models. As stated in the literature, the latter two models were linearized before fitting to avoid the complexity associated with their nonlinear counterparts. For example, the code for the Hixson–Crowell model in MATLAB generates columns with Y = (1 − *CFR*)^1/3^ as follows:

for i = 1:length(ydata)newY(i) = (1 − ydata(i))^(1/3);end[est,gof]=HixsonC(xdata, newY.’);coeff = coeffvalues(est);ylbl = ‘(1 − *CFR*)^(1/3)’;

Subsequently, the MATLAB built-in functions for regression were implemented, as shown below. The nonlinear least squares method was selected as the regression method; the constant(s) to be fitted were constrained by lower and upper bounds, and initial guesses were provided. The output of the ‘fit’ command returns the estimated values of the model coefficient(s) and the goodness-of-fit statistics as a table that includes the *SSE* values.

function [est,gof] = HixsonC(X,Y)s = fitoptions(‘Normalize’,‘off’,…,‘Method’,‘NonlinearLeastSquares’,‘Lower’,[−100],‘Upper’,[100],…‘StartPoint’,[1]);f = fittype(‘1 − a*x’,‘independent’,{‘x’},‘coefficients’,{‘a’},‘options’,s);[est,gof] = fit(X,Y,f);end

After executing the MATLAB code, a dialog box is displayed to prompt the user to select a single or several models for data fitting; then data for the nine replicates in the spreadsheet will be processed based on the model selection. A MATLAB code for an output generates a report in word document (.docx) format displaying the following information (Figure 6):○Estimated model coefficient(s) for each run.○*SSE* values for each run with a statistical summary.○Plots of *CFR* vs. time for each run based on the selected model.○A summary page, including a summary plot of all the moieties and power densities in the spreadsheet.

Another output report is generated similar to the one above but in EXCEL (.xls) format.

## 4. Conclusions

Liposomes are promising DDSs. These nanosized vehicles can be tailored to target specific receptors overexpressed on the surface of tumor cells by conjugating suitable ligands to their surface. They can also be designed to release their cargo in a sustained manner in response to internal or external stimuli. In this work, in vitro drug release profiles of sonosensitive liposomes functionalized with seven different ligands were fitted to kinetic models to find the best fit(s). This will aid in predicting drug release profiles for future studies. Low- and high-frequency ultrasound were used to induce drug release with several power densities. At a low frequency (i.e., 20 kHz), Hixson–Crowell, Korsmeyer–Peppas, Weibull, Gompertz, and Lu–Hagen showed good fits to our release profiles at all three power densities. At high frequencies (1.07 and 3.00 MHz), the Hixson–Crowell, Korsmeyer–Peppas, and Gompertz were the best-fitting models. In addition, a third-order polynomial reflected the best fitting at all frequencies and power densities, albeit such a model lacks a physical representation/meaning.

The sonosensitivity of liposomes is affected by many factors other than the acoustic parameters (i.e., frequency, power density, and pulse duration). The physical properties of liposomes, such as the nature of the lipids, targeting moieties, and the medium of release, may also influence the liposomal response to ultrasound. Future work includes formulating an empirical equation that relates the molecular weight (MW) and the acid dissociation constant (pKa) of conjugated moieties to release constants (k_i_) predicted for single-coefficient models such as the zero-order and Korsmeyer–Peppas.

## Figures and Tables

**Figure 1 biomedicines-10-03139-f001:**
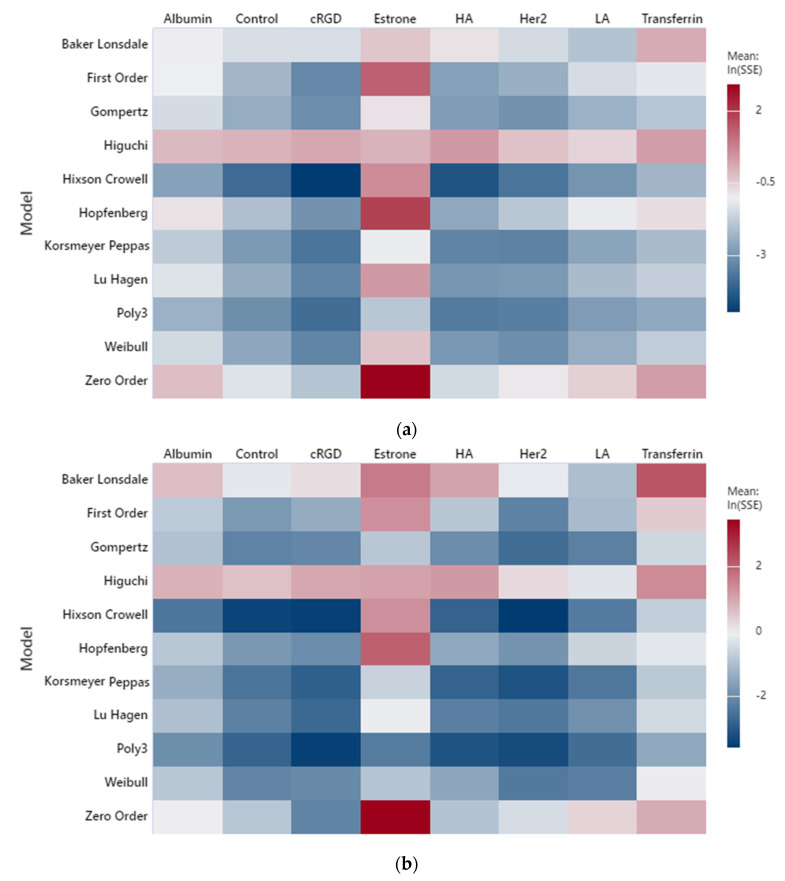
Heatmap for LFUS fittings at (**a**) 6.2 mW/cm^2^, (**b**) 9 mW/cm^2^ and (**c**) 10 mW/cm^2^.

**Figure 2 biomedicines-10-03139-f002:**
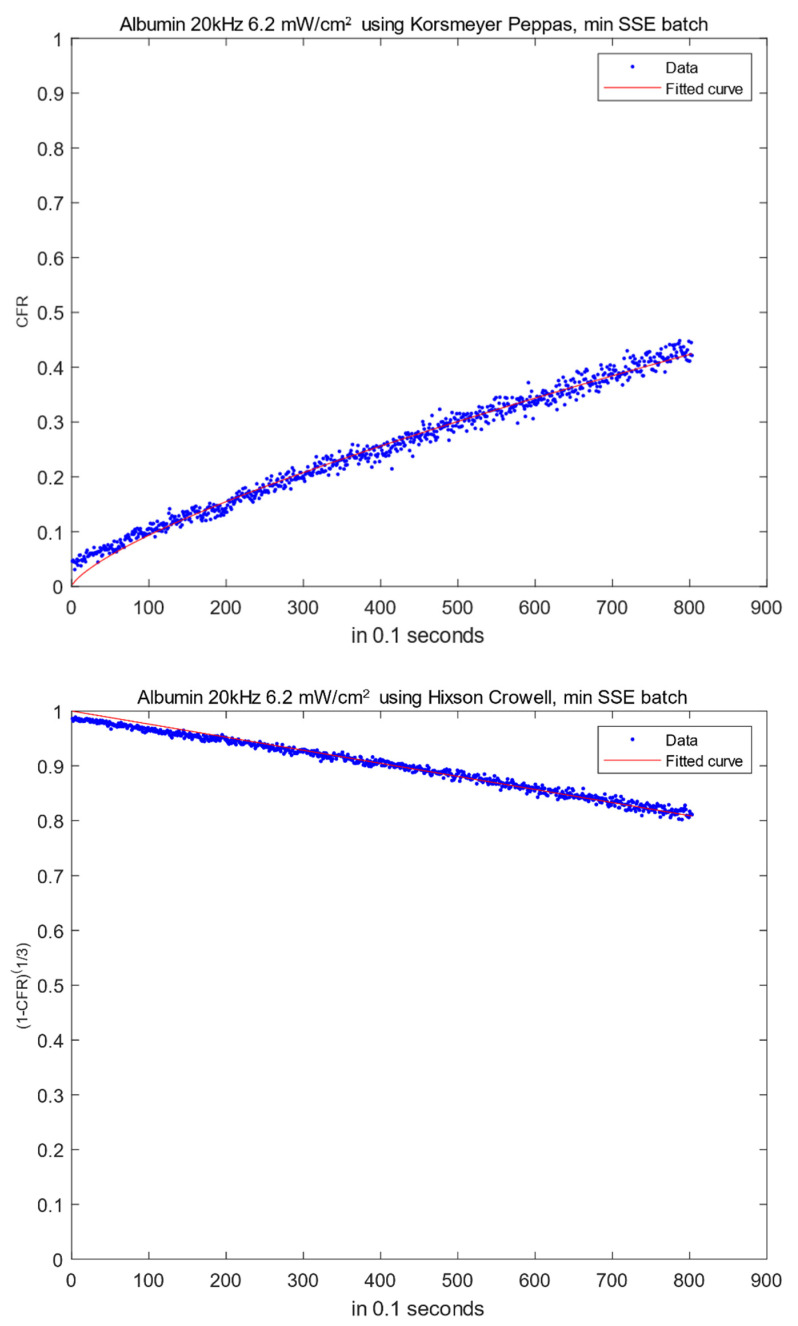
Selected model fittings for albumin at 20 kHz (6.2 mW/cm^2^).

**Figure 3 biomedicines-10-03139-f003:**
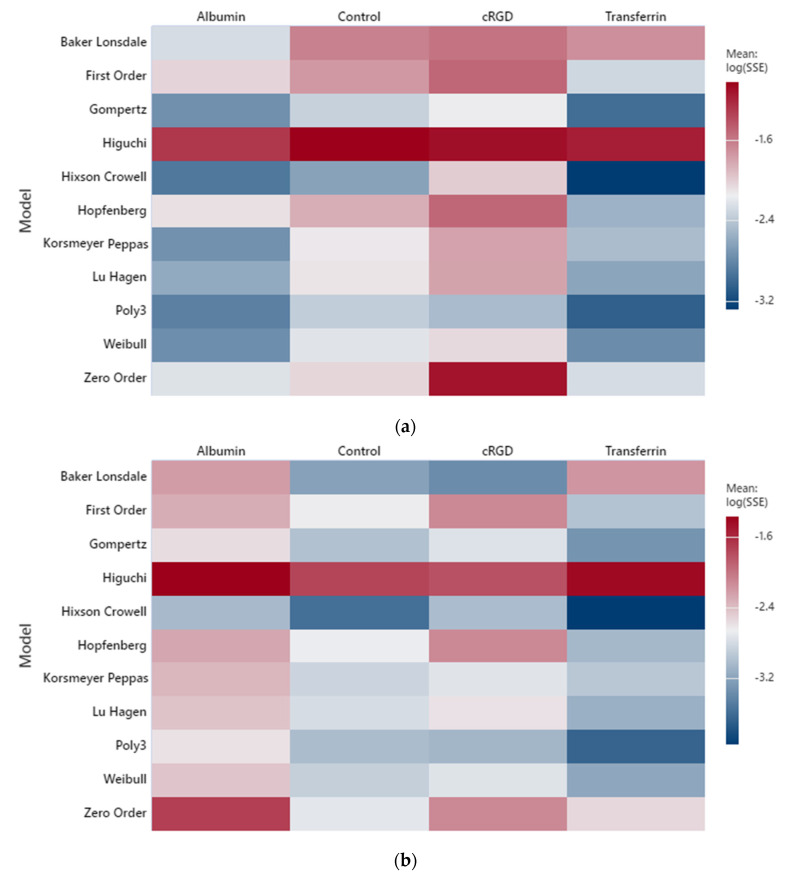
Heatmap for HFUS fittings at (**a**) 1.07 MHz (10.5 W/cm^2^) and (**b**) 3.00 MHz (173 W/cm^2^).

**Figure 4 biomedicines-10-03139-f004:**
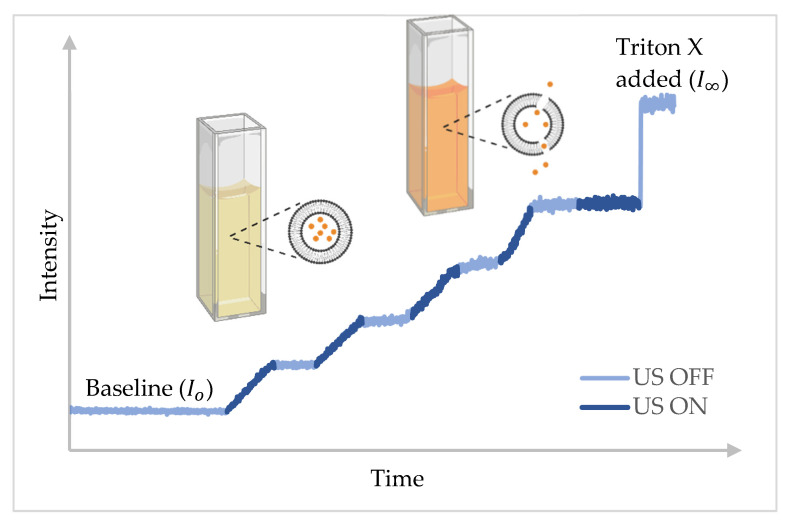
Calcein release from calcein-loaded liposomes using pulsed LFUS.

**Figure 5 biomedicines-10-03139-f005:**
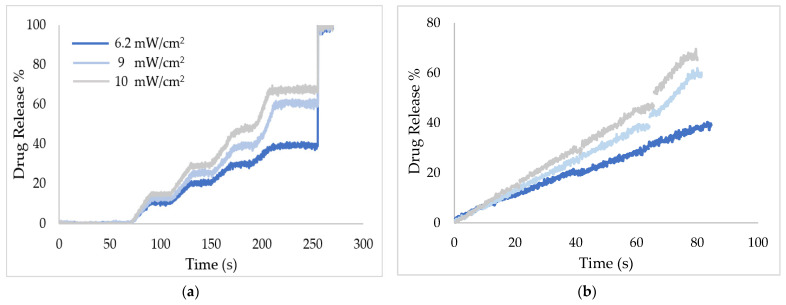
LFUS release profiles for the cRGD-conjugated liposomes. (**a**) Stepped and (**b**) de-stepped data.

**Figure 6 biomedicines-10-03139-f006:**
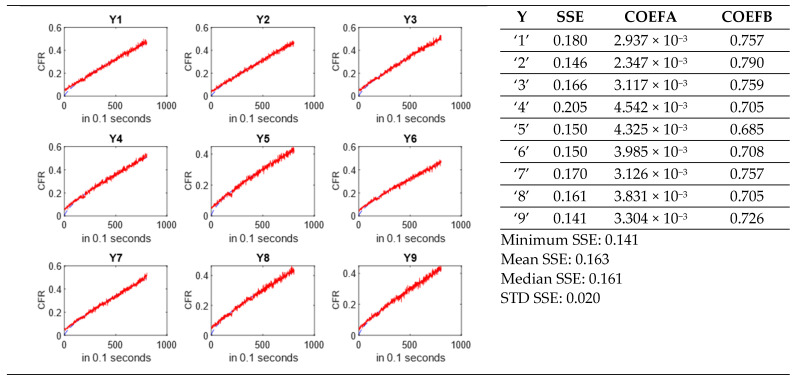
Output report displaying *SSE* values and model coefficients for albumin at 20 kHz (6.2 mW/cm^2^) using Korsmeyer–Peppas.

**Table 1 biomedicines-10-03139-t001:** The advantages and disadvantages of drug delivery trigger mechanisms.

Type	Advantages	Disadvantages
visible/near-infrared light	▪ very precise ▪ inexpensive ▪ minimally invasive and provides functional information ▪ non-ionizing radiation	▪ limited tissue penetration (1–10 cm)
pH	▪ wide applicability ▪ simple structure	▪ low accuracy (pH may vary depending on the severity of disease or proximity to diseased tissue) ▪ maintaining structure during the process of delivery may be challenging ▪ pH-sensitive systems are susceptible to off-target delivery
magnetic field	▪ energy modulation with an alternating magnetic field (AMF) ▪ provides imaging opportunity ▪ no limit on tissue penetration ▪ non-ionizing radiation	▪ expensive ▪ limited to surface tumors ▪ accumulation of particles within a magnetic field can lead to embolism and/or increased cytotoxicity
ultrasound	▪ non-invasive ▪ low cost ▪ fast ▪ easily accessible ▪ spatiotemporal control ▪ high patient acceptability and synergism with therapeutic agents ▪ non-ionizing radiation	▪ difficult to apply homogeneously to large volumes ▪ can lead to temperature rise
microwave	▪ non-invasive ▪ easy to generate and control ▪ non-ionizing radiation	▪ low penetration ▪ can lead to temperature rise

**Table 2 biomedicines-10-03139-t002:** Targeting moieties general information.

Targeting Moiety	Tumor-Overexpressed Receptor(s)	Reference
estrone (ES)	ER+ breast cancer	[25,26]
albumin	Gp60, SPARC and hnRNPs overexpressed in MDA-MB-453 breast cancer, lung cancer, metastatic pancreatic adenocarcinoma, and melanoma	[18,27]
cyclic arginylglycylaspartic acid (cRGD)	αvβ3 integrin overexpressed in pancreatic, renal, and breast cancer	[28,29]
herceptin (HER2)	HER2 overexpressed in breast, ovarian, gastric, and prostate cancer	[20,30]
hyaluronic acid (HA)	CD44 overexpressed in breast, lung, colorectal, gastric, renal hepatocellular, pancreatic, cervical cancers, and melanomas	[21,31]
lactobionic acid (LA)	ASGRP overexpressed in liver cancer	[23,32]
transferrin (T_f_)	T_f_ receptors overexpressed on all the rapidly dividing normal and malignant cells such as liver, cervical and ovarian cancers, leukemia, neuroblastoma, and glioblastoma	[33,34]

**Table 3 biomedicines-10-03139-t003:** Kinetic release models from [35,36,37].

Model	Model Expression ^1^	Modified Expression	Conditions and Uses
zero-order	$Q_{t}-Q_{0}=k_{0}t$	$CFR=\frac{Q_{t}-Q_{0}}{Q_{\infty }}=k_{0}ft$	▪ concentration-independent (constant release) ▪ osmotic systems, transdermal systems and tablets encapsulating agents of low solubility
first-order	$\ln C_{t}=\ln C_{0}-k_{1}t$	$CFR=\frac{C_{t}-C_{0}}{C_{T}}=\frac{C_{0}}{C_{T}}\left(e^{-k_{1}t}-1\right)$	▪ concentration-dependent release (1st order) ▪ soluble agents encapsulated in porous systems
Higuchi	$Q_{t}=A\sqrt{DC_{s}\left(2C_{lip}-C_{s}\right)t}$	$CFR=k_{h}\sqrt{t}$	▪ water-soluble drugs in semisolid and solid matrices ▪ initial drug concentration is very high in the matrix compared to the drug solubility ▪ drug molecules are relatively small in size with respect to the DDS thickness ▪ one-dimensional diffusion with constant diffusivity of the drug ▪ negligible volume change of the DDS ▪ perfect sink in the release medium
Hixson–Crowell	$W_{0}^{1/3}-W_{t}^{1/3}=\frac{DK^{\prime }C_{s}N^{1/3}}{\delta }t$	${\left(1-CFR\right)}^{1/3}=1-k_{hc}t$	▪ valid in drug systems with diminishing surface area and diameter ▪ used to model tablets when dissolution occurs in planes parallel to the surface of the drug but maintaining the geometrical characteristics
Korsmeyer–Peppas (power law)	$\frac{Q_{t}}{Q_{\infty }}=k_{kp}t^{n}$	$CFR=k_{kp}t^{n}$	▪ semi-empirical equation assuming no drug release initially ▪ used mainly in polymeric systems such as hydrogels ▪ the mechanism of drug release is reflected in the value of exponent n
Baker–Lonsdale	$\frac{3}{2}\left[1-{\left(1-\frac{Q_{t}}{Q_{\infty }}\right)}^{2/3}\right]-\frac{Q_{t}}{Q_{\infty }}=\frac{3DC_{ms}}{r_{0}^{2}C_{0}}t$	$\frac{3}{2}\left[1-{\left(1-CFR\right)}^{2/3}\right]-CFR=k_{bl}t$	▪ negligible drug release initially ▪ homogenous non-fractured spherical matrices
Weibull	$Q_{t}=Q_{\infty }\left(1-e^{-k^{\prime }t^{b}}\right)$	$CFR=1-e^{-at^{b}}$	▪ empirical model ▪ useful in comparing drug release profiles for matrix-type systems
Gompertz	$X_{t}=aX_{max}e^{-be^{clog\left(t\right)}}$	$CFR=\frac{X_{t}}{X_{max}}=k_{g}e^{-ce^{blog\left(t\right)}}$	▪ useful in comparing release profiles of soluble drugs with an intermediate release rate
Hopfenberg	$\frac{Q_{t}}{Q_{\infty }}=1-{\left[1-\frac{k_{er}t}{C_{0}a_{0}}\right]}^{b}$	$CFR=1-{\left[1-k_{hf}t\right]}^{b}$	▪ used for drug release from surface-eroding polymers with constant surface area during degradation
Lu–Hagen	$\frac{Q_{t}}{Q_{0}}=k_{lh1}-\frac{k_{lh1}}{1+k_{lh2}t}$	$CFR=\frac{k_{lh1}k_{lh2}\;t}{1+k_{lh2}t}$	▪ developed for the rapid release of drugs from thermosensitive DDSs with a size below 100 nm ▪ Laplace pressure is the driving force for release at the optimum temperature rather than the concentration gradient (i.e., diffusion)

^1^ Refer to the Appendix A for the glossary of symbols.

**Table 4 biomedicines-10-03139-t004:** Korsmeyer–Peppas model coefficients a and b at 20 kHz.

Moiety	Power Density (mW/cm^2^)					
	6.2		9		10	
	*a*	*b*	*a*	*b*	*a*	*b*
albumin	3.50 × 10^−3^	0.73	3.25 × 10^−3^	0.79	5.70 × 10^−3^	0.74
cRGD	1.32 × 10^−3^	0.86	9.83 × 10^−4^	0.95	9.40 × 10^−4^	0.99
estrone	5.33 × 10^−2^	0.41	8.32 × 10^−2^	0.36	1.11 × 10^−1^	0.32
HA	1.67 × 10^−3^	0.85	2.52 × 10^−3^	0.85	4.78 × 10^−3^	0.78
Her2	3.48 × 10^−3^	0.77	4.46 × 10^−3^	0.77	3.35 × 10^−3^	0.87
LA	3.33 × 10^−3^	0.72	5.57 × 10^−3^	0.69	9.45 × 10^−3^	0.64
transferrin	5.61 × 10^−3^	0.73	6.64 × 10^−3^	0.75	1.70 × 10^−2^	0.62

## Data Availability

Not applicable.

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
