# Peer review of "Modeling of the In Vitro Release Kinetics of Sonosensitive Targeted Liposomes"

_biomedicines, 2022, doi:10.3390/biomedicines10123139_

Round 1

Reviewer 1 Report

In the manuscript titled "Modelling of the in vitro release kinetics of sonosensitive targeted liposomes", in vitro drug release profiles of sonosensitive liposomes functionalized with 7 different ligands were fitted to kinetic models to find the best fit(s). Low- and high-frequency ultrasound were used to induce drug release with several power densities. This work is well developed and supported by the results, and the authors have done adequate data analysis. These models can help others in predicting drug release profiles. After addressing the following questions properly, the manuscript could be considered for publication.

1. The material part for the chemicals used in this manuscript is required.

2. Is the Doxorubicin used in this work Doxorubicin free base or Doxorubicin hydrochloride?

3. I understand that this work aims to examine the kinetics of drug release from sonosensitive liposomes with several targeting moieties based on data gathered by references from 9 to 18. Could the authors summarize the methods for preparing the liposomes used in this work and their related characterization (such as size, PDI, and TEM images)? Some readers may not have access to all these 10 papers, so it will be helpful to include the necessary information on liposomes in this manuscript.

4. L44-46: The authors are asked to introduce every acronym before using it in the text. The first time you use the term, put the acronym in parentheses after the full term, for instance, “Capecitabine and oxaliplatin (CAPOX) and folinic acid, fluorouracil, and oxaliplatin (FOLFOX) for colon cancer and others.”

Reviewer 2 Report

Few questions/comments

1. Please add some references about DDS.

2. What is SSE (line 88)? Reference, or decoding.

3. Table 3 - please define "a" and "b" parameters

4. Line 127-129  - how you can interpret this statistical result?

5. There are alot of abbreviations in the text, some of which are deciphered, and some are not. And also - some methods have reference, and some not.

Reviewer 3 Report

The research article “Modeling of the in vitro Release Kinetics of Sonosensitive Targeted Liposomes” is a well written and scientifically sound piece of work. There are few corrections that need to be made to the final manuscript.

1.      Line 31: The authors mentioned that chemotherapy is commonly used either as a primary treatment or combined with other treatments such as adjuvant therapy and neoadjuvant therapy. Consider rewriting the sentence as adjuvant and neoadjuvant therapy also reflects the chemotherapy.

2.      Consider giving the full form of the abbreviations/acronyms first time you use the term in the manuscript. For example, cRGD, MOPP, CHOP, CAPOX, FOLFOX, AMF, LF, US, Dox etc.

3.      Consider providing some introduction about the modeling in the section 1?

4.      What is the pH of PBS used for the experiments?

5.      Why the authors applied two different ultrasound ON/OFF pulsed modes for calcein-loaded liposomes and Dox-loaded liposomes? Is there any specific reason for it?

6.      There are few typographical errors which need to be corrected.
